# Sonographic Assessment of Hyperechoic Vertical Artifact Characteristics in Lung Ultrasound Using Microconvex, Phased Array, and Linear Transducers

**DOI:** 10.3390/vetsci12100949

**Published:** 2025-10-01

**Authors:** Michał Gajewski, Katarzyna Kraszewska, Kris Gommeren, Søren Boysen

**Affiliations:** 1Vetcardia Veterinary Clinic Kijowska 11, 03-743 Warsaw, Poland; vetlusexpert@gmail.com; 2Department of Clinical Sciences, Faculty of Veterinary Medicine, Fundamental and Applied Research for Animals & Health, University of Liège, 4000 Liège, Belgium; kris.gommeren@uliege.be; 3Faculty of Veterinary Medicine, University of Calgary, Calgary Alberta, T2N 1N4, Canada; srboysen@ucalgary.ca

**Keywords:** lung ultrasound (LUS), hyperechoic vertical artifacts, B-lines, POCUS, microconvex transducer, linear transducer, phased-array transducer, dogs

## Abstract

**Simple Summary:**

Lung ultrasound is a widely used, non-invasive tool in both human and veterinary medicine. One of its key features is the appearance of bright vertical artifacts, which can provide important information about lung health. However, these artifacts can appear differently depending on the type of ultrasound transducer used, which may make diagnosis more challenging. This study compared three different types of transducers in dogs to see how well each one displayed these vertical artifacts. We found that overall image quality and the ease of evaluating the artifacts varied between transducers. The transducer commonly used in cardiology produced lower-quality images, making it harder to see and count the artifacts clearly. In contrast, the other two transducers generally gave clearer and more reliable images. These results show that the choice of transducer can strongly influence how lung changes are detected in dogs. A better understanding of which transducers provide the clearest images will help clinicians make more accurate diagnoses, which ultimately improves care for animal patients.

**Abstract:**

Hyperechoic vertical artifacts are an essential feature of lung ultrasound (LUS) arising from various pathological states. Those that meet the criteria for B-lines have the most significant diagnostic value and should be differentiated from other hyperechoic vertical artifacts of unspecified clinical importance. Although numerous studies have assessed the impacts of transducer type on the appearance of B-lines in human medicine, comparative studies in veterinary medicine are limited and conflicting. This study compares three transducer types for the assessment of hyperechoic vertical artifacts in dogs. We hypothesize that there is high-level reviewer agreement in the assessment of HVA image quality and characteristics, and that the image quality/characteristics differ between the three transducers. Dogs (n = 8) with HVAs and sonographic absence of lung consolidations, pleural effusion, and/or pneumothorax were enrolled. Twenty-four cine-loops (5 s) containing HVAs were retrospectively and independently reviewed by two reviewers, who were blinded to the case details but not transducer type. The reviewers assessed the cine-loops for the following: whether HVAs meet the B-line criteria, ease of counting HVAs, and overall image quality. Paired cine-loops from the same patient using different transducers were then presented for HVA quality comparison. Inter-rater concordance was determined using the Kappa coefficient, Kendall’s tau, and Pearson correlation coefficient, while characteristics were compared using chi-square and Kruskal–Wallis tests (level of significance, α = 0.05). The overall concordance of image quality was good (Pearson’s coefficient = 0.82). The PA transducer scored lower in image quality (*p* < 0.001), HVA blending (*p* = 0.014), graininess (*p* < 0.001), and clarity of edges (*p* < 0.001) when compared with the microconvex and linear transducers, and the identification of B-line criteria differed between transducers (*p* = 0.024). Furthermore, the PA scored lowest in the comparison of paired cine-loops regarding the image and HVA quality (*p* < 0.001). Although more HVAs failed to reach the far field with the linear transducer (10/16, 62.5%) compared with the microconvex (8/16, 50%) and PA (3/16, 18.5%) transducers, the linear transducer scored higher than the microconvex and PA transducers regarding its ability to count B-lines (*p* < 0.001). This study demonstrates that the type of transducer significantly impacts the characteristics of HVAs, with the PA transducer producing lower-quality images compared with the microconvex and linear transducers.

## 1. Introduction

Lung ultrasound (LUS) is an imaging modality utilized in human and veterinary medicine to assess various pathologic processes involving the lungs, pleura, and pleural space [1,2]. Although various studies have investigated the effects of transducer type on the quality of images obtained when assessing normal and pathologic lungs and pleura in human medicine, knowledge regarding the impacts of transducer type on the identification of normal and abnormal lung and pleural characteristics in veterinary medicine remains limited [3,4,5,6,7]. Recent consensus statements in human medicine recommend the use of a curvilinear transducer for sonographic assessment of the lung and pleura of adults [8,9]; however, in neonatology, the use of a linear transducer is recommended for this purpose [10,11]. It is unknown whether either of these transducer recommendations should be applied to adult companion animals.

This study compares three transducer types (microconvex, linear, and phased array [PA]) for the assessment of hyperechoic vertical artifacts (HVAs) in dogs with diseased lung states. We hypothesize that the image quality/characteristics differ between the three transducers, despite a high level of agreement between reviewers regarding the image quality and HVA characteristics. The secondary goals of the study were to identify the most reliable transducer for assessing HVAs in patients presenting with lung pathologies and to calculate the inter-rater concordance when assessing cine-loops to define the most reproducible quality assessment parameters.

## 2. Materials and Methods

### 2.1. Animals

Eight client-owned dogs that presented to a private clinic for cardiac consultation and an echocardiographic exam were prospectively enrolled after receiving verbal consent from their owners. According to the opinion of the local ethical committee in Warsaw, no formal ethics consent was required for this study except for the informed consent of participants. Inclusion criteria were the presence of HVAs on LUS, regardless of etiology. Exclusion criteria included lung consolidation (defined as the presence of a single, large hypoechoic area of subpleural non-aerated lung visible on any transducer), large quantities of pleural effusion (defined as any amount of fluid visible on any transducer causing separation of the visceral and parietal pleura), and/or pneumothorax (defined by lack of lung sliding and an absence of B-lines on LUS) diagnosed on LUS.

### 2.2. Lung Ultrasound Examination

The examination was performed following a predefined standardized approach by one of two experienced sonographers with over 5 years of experience in LUS (MG and KK). The LUS technique was performed according to a sliding protocol described elsewhere [12,13]. Briefly, a horizontal sliding technique with the transducer placed at three different vertical locations on each side of the thorax was utilized. The starting orientation of the probe is perpendicular to the ribs. The transducer is then slid along a dorsal line from cranial to caudal, rotated 90 degrees into a transverse plane (parallel to the ribs), and slid again from caudal to cranial direction along the same line. This process is repeated in the middle and ventral thirds of the hemithorax and then repeated on the contralateral hemithorax. All LUS exams were performed with a 3–8 MHz microconvex transducer, a 3–6 MHz phased array (PA) transducer, and a 6–12 MHz linear transducer, and cine-loops were recorded using a GE Vivid IQ ultrasound machine. The transducers were always held in the same position relative to the patient, perpendicular to the ribs. When a site positive for HVAs was identified with a microconvex transducer, 3–5 s cine-loops were recorded with imaging depth set at 8 to 10 cm. The operator fixed their hand on the lateral thoracic wall of the patient and then replaced the transducer without moving their hand. Using this technique, cine-loops were recorded for the same thoracic window with the PA transducer, followed by the linear transducer (Figure 1).

The ultrasound settings were chosen based on previous publications [4,14], which have stated the need for a specific lung ultrasound preset to obtain images with the least impact of machine post-processing on naturally occurring artifacts. For this purpose, harmonics should be turned off, the frequency set to the lowest value for the transducer, persistence set to zero, the focal point set at the level of the pleural line, and time-gain compensation (TGC) increased at the distal (far) field of the screen. These settings were utilized in the current study, with the exception that the TGC was not increased in the far field; instead, the TGC was set to a mid-range level throughout to create a more uniform echogenicity between images and transducers, facilitating comparison by the evaluators [4,14,15,16,17]. Cine-loops (n = 24) were stored for retrospective evaluation by two blinded reviewers.

### 2.3. Evaluation

Two board-certified emergency and critical care specialists, each with more than 10 years of research and clinical experience in lung ultrasound and POCUS (SB and KG), who were blinded to the patients’ history and clinical diagnosis, but not the type of transducer used, evaluated all cine-loops. Before reviewing, the evaluators agreed on the characteristics of the HVAs they would interpret and pre-assessed several cine-loops to ensure that they had a similar understanding of the characteristics being evaluated. The clips were randomly presented as individual clips to the evaluators, who assessed the image quality based on the HVA characteristics (Table 1). The examiners were asked to fill out a questionnaire including closed questions regarding the quality of vertical artifacts, scoring on a Likert scale for subjective assessment of images, and open questions to explain their reasons for rating the quality of images obtained using specific transducers (see Table 1). Following the assessment of individual clips, sets of paired cine-loops (two transducers from the same window of each patient, i.e., microconvex vs. PA, microconvex vs. linear, and PA vs. linear) were directly compared regarding the image quality. The reviewers compared the overall image quality (equal vs. unequal) and, if unequal, selected the transducer cine-loop they felt had the higher HVA image quality, providing an explanation (from a limited number of choices) for why they believed that one transducer had a higher image quality than the other (see Table 1). The scores for each cine-loop and each pair of clips were recorded for further statistical analysis.

### 2.4. Statistical Analysis

Statistical analyses were performed using IBM SPSS Statistics 30. Analysis of concordance was carried out using the Kappa coefficient (for nominal variables), Kendall’s tau (for ordinal variables), and Pearson’s tau (for quantitative variables). Furthermore, a series of chi-square tests of independence, for which frequencies are reported, and Kruskal–Wallis tests, for which means and standard deviations are reported, were performed. The level of significance was set at α = 0.05. The concordance of the evaluators was further analyzed using Krippendorff’s α coefficient to verify the overall agreement between the experts regarding the parameters assessed on nominal, ordinal, and quantitative scales. The pairwise comparisons of transducers were analyzed through a univariate analysis of variance and a Games–Howell post hoc test. A total of 16 cine-loops from each transducer were assessed, and each transducer, regarding each clip and each parameter, was assigned a score ranging from 0 to 2 points. In the case that two transducers were assessed equally, 0.5 points were assigned to each of them; meanwhile, when all three transducers were assessed equally, 1 point was assigned to each.

## 3. Results

The concordance results of the two reviewers regarding the individual transducer characteristics are presented in Table 2. Assessments of image and HVA quality using nominal scales are generally subjective, and, accordingly, the analysis revealed that the level of agreement was moderate for variables evaluated on a nominal scale (see Table 2), such as the graininess of HVAs and their tendency to blend or have unclear margins, variable width, or move too quickly. On the other hand, the evaluators showed high concordance when assessing parameters on the ordinal and quantitative scales (see Table 2). Krippendorff’s α coefficient was used to verify the overall agreement between the experts for parameters assessed on nominal, ordinal, and quantitative scales. The results indicated that the agreement for the 23 parameters assessed on a nominal scale was moderate (α = 0.68; 95% CI: LL = 0.61; UL = 0.75); the agreement for the 3 parameters assessed on an ordinal scale was good (α = 0.79; 95% CI: LL = 0.65; UL = 0.90); and the agreement for the 3 parameters assessed on a quantitative scale was very good (α = 0.95; 95% CI: LL = 0.88; UL = 0.99).

The diagnostic characteristics for each transducer are presented in Table 3, while the results for the paired transducer cine-loops comparisons (i.e., microconvex vs. PA, microconvex vs. linear, and PA vs. linear) are presented in Table 4. The analysis showed statistically significant differences between the compared transducers regarding all the parameters. Moreover, it is worth noting that all the observed effects were strong (η^2^ > 0.14). The obtained results indicate that regarding image quality parameters, the linear transducer scored higher than both the microconvex and PA transducers, and the microconvex transducer scored higher than the PA transducer. The HVA movement was easier to assess when using the microconvex transducer, compared with the PA transducer.

## 4. Discussion

The results of this study confirm the hypothesis that the type of transducer used in lung ultrasound influences both image quality and the appearance of HVAs in dogs. Importantly, understanding the features of HVAs and image quality associated with each of the transducers may help clinicians to avoid misinterpreting the results. For example, the PA transducer performed poorly regarding quantification of HVAs, which may impact serial measurements of B-lines over time or result in different findings between operators. The poor quantification of HVAs using the PA transducer was considered to be a result of poorly defined edges and increased graininess in the appearance of HVAs. The overall image quality score (on a Likert scale) for individual clips was also lower for the PA transducer when compared with the microconvex and linear transducers, with good inter-rater agreement. In the final part of the analysis, the quality of images and HVAs was assessed for paired cine-loops from different transducers. These results indicated a significant difference in the quality of the image when the cine-loops were compared pairwise rather than being assessed alone. The comparison showed that, based on the reviewers’ image quality assessments, the PA transducer was perceived as inferior to the microconvex and linear transducers, with the PA transducer never considered to have a higher image quality than any other transducer in any of the patients by either of the evaluators. It is also worth noting that the linear transducer was perceived to be superior to the microconvex and PA transducers regarding visualizing the quality of the pleural line, even though the cine-loops were recorded at a depth of 8 to 10 cm. This finding may support the conclusions of a recent study suggesting that using a high-frequency linear transducer may result in improved diagnostic accuracy compared with a microconvex transducer [19].

According to data from the literature, HVA characteristics can vary with machine settings [4,14,16,17]. The machine settings used in the current study were standardized for all transducers to minimize any variability in image quality and HVA characteristics. In previously published veterinary studies, the specific machine settings for LUS are usually reduced to stating the transducer type, frequency, and imaging depth [2,5,19,20], with additional information on focus position [3] and harmonics [5] seldom provided. Whether this lack of information and potential uniformity in machine settings could have an impact on the discrepancies between the results of these studies remains unknown. However, based on research on the physics of ultrasound in LUS, it can be concluded that the machine settings (central frequency, transducer frequency range, focus position, and TGC) influence the appearance of artifacts [4,21]. The justification for setting the lowest frequency for each transducer is the previously reported phenomenon of shortening of HVA length when using a higher frequency [18]. In our study, the TGC was set to a mid-range level throughout the imaging depth, mimicking a busy general practice or emergency clinical setting where there may not be time to adjust the TGC scale when performing rapid sonographic lung evaluations in dyspneic patients. Furthermore, in the GE Vivid IQ machine, the TGS is set in the same manner for the microconvex and PA transducers, and our intention was to utilize the same machine settings for all transducers. It is possible that the appearance and length of the HVAs would have changed—and, consequently, the expert assessments may have been different—had the TGC been optimized throughout the field of vision for each probe and for each animal.

Hyperechoic vertical artifacts are created in lung ultrasound by various acoustic traps of different widths, depths, and positions, depending on the underlying lung pathology [21,22]. Several mechanisms, physical laws, and mathematical principles have been proposed to explain the formation of such artifacts [23,24]. Interestingly, depending on the type of acoustic trap and the width of the trap’s access channel, the resulting vertical artifacts can exhibit different properties, i.e., they can be modulated or unmodulated, and may have different widths, echogenicity, and lengths [21,22,25,26]. One of the methodological challenges of this study was identifying uniform criteria to define HVAs and B-line artifacts. Criteria used to define “true” B-lines in both the human and veterinary literature are conflicting. To further complicate the issue, there is a lack of consensus on how to differentiate B-lines, Z-lines, and I-lines [21]; for example, some authors have reported that shorter HVAs such as I-lines should move with lung sliding, while Z-lines should not, which helps to distinguish them from B-lines [27]. In contrast, other authors have stated the opposite: that Z-lines move with lung sliding and I-lines do not [15,21]. Referring to publications on the physics of ultrasound in LUS, the characteristics in question are influenced by the type of acoustic trap; as such, it is possible that the channel connecting the pleural surface to the interior of the trap is so narrow that it allows only minimal transmission of pulse energy [21,25]. Theoretically, attenuation may also contribute to the short length of an artifact [21]. Due to the size of the acoustic trap and the length of the connecting channel, a B-line often originates slightly below the pleural plane [21,23]. From a clinical perspective, however, the most important issue is to distinguish “true” B-lines from other HVAs (regardless of the adopted nomenclature), as B-lines are considered pathological artifacts whereas other HVAs—depending on the author—are reported to have variable clinical significance and lack a clearly defined pathological basis.

For this study, an HVA was considered a true B-line if it met the following criteria: it arises from the pleural line, moves with lung sliding, and extends to the bottom of the screen irrespective of imaging depth. Under these assumptions, the study revealed that most HVAs met the B-line criteria more frequently when using the PA transducer than the linear transducer, while the microconvex transducer did not differ in this respect from the other two. The most common reason for HVAs not meeting the B-line criteria for the linear transducer was that they did not extend to the bottom of the screen, with the number of such HVAs being higher for the linear transducer than for the other two transducers. However, the HVAs were considered significantly easier to count with the linear transducer than with the microconvex and PA transducers; in particular, compared with the linear transducer, it was harder to clearly distinguish individual HVAs when using the PA transducer. Furthermore, there was no consensus between the evaluators regarding whether HVAs originated from the pleural line, and only limited agreement on whether they extended to the bottom of the screen. These inconsistencies may have led to the moderate level of agreement regarding the fulfillment of B-line criteria. This finding underscores that B-line assessment is more complex than suggested in prior studies. In contrast, the inter-rater agreement was high for quantitative parameters, including whether HVAs did or did not reach the bottom of the screen and overall image quality. These results indicate that structured numerical scales, such as the LUSS or Likert-based measures with predefined criteria, may reduce the inter-rater variability, whereas subjective parameters remain less reliable.

This study has several limitations. It was not possible to blind the reviewers to the transducer type, which may have created a pre-existing preference bias between them, and prior familiarity with one type of transducer over another may have also led to greater comfort in interpreting HVAs. Although the placement of each transducer on the thorax was controlled, it is possible that the site was not identical between transducers or that slight displacement of the examiner’s hand or patient movements may have influenced the findings and image quality. Furthermore, both reviewers were considered highly experienced evaluators, and the results may not apply to less experienced sonographers; however, the sensitivity of LUS increases with expertise, and thus, it is questionable whether responses from less experienced evaluators would contribute to the true quality assessment. Although the inclusion of more evaluators may have increased the robustness of the study, given the significant difference between the PA transducer and the other two transducers when comparing paired cine-loops, it is unlikely that this tendency would change with an increased number of evaluators. The study population was small and focused on dogs presenting for cardiac consultation, and it is unclear whether the same results would be obtained when using a larger group and/or including dogs with a wider range of underlying lung pathologies. The TGC was not adjusted to the depth of imaging on the linear probe, which may have impacted the visualization of HVAs (as discussed above). All examinations were performed with only one type of ultrasound machine, and image quality may differ on other machines. Finally, the evaluation of the cine-loops was performed retrospectively, and the perceived image quality may have differed if the assessment was performed in real-time. The quality loss due to the transfer and conversion of files was probably minimal and equal across all cine-loops, given that only one machine was used for all image captures.

## 5. Conclusions

The transducer used to assess the lungs can have a clinically significant impact on the interpretation of HVAs. The linear transducer led to the highest image quality score, with the microconvex transducer scoring slightly lower; meanwhile, both of these transducers outscored the PA transducer. Pleural line abnormalities were best visualized with the linear transducer. However, more HVAs failed to reach the far field of the screen with the linear transducer than the other transducers, which may result in misclassification of B-lines for other HVAs of unspecified clinical importance by less experienced sonographers. More HVAs met the criteria used to define a B-line when using the PA transducer; however, this transducer performed poorly regarding quantifying HVAs, which may impact the serial measurement of B-lines over time or result in different findings between operators. It is therefore possible that the severity of lung injury may be overestimated when using a PA transducer, due to the blending of HVAs and poor overall image quality. Incorporating a numerical scale (such as the LUSS or a Likert scale) into LUS examinations may decrease inter-rater variability. Further research is needed to develop clear definitions of various HVAs and assess their importance in the context of small animal LUS and POCUS examinations.

## Figures and Tables

**Figure 1 vetsci-12-00949-f001:**
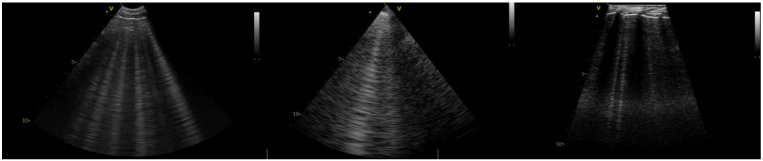
Still images from cine-loops recorded with three transducer types (from left to right): microconvex, PA, and linear.

**Table 1 vetsci-12-00949-t001:** Questions regarding the quality of hyperechoic vertical artifacts (HVAs) assessed by the evaluators.

Questions	Answers
Part 1: the assessment of individual clips for image quality	
1. Are hyperechoic vertical artifacts (HVAs) visible?	Yes/No
2. If yes, do HVAs meet the criteria of B-lines? *	Yes, all HVAs meet the criteria Yes, but only some HVAs meet the criteria No; none of HVAs meet the criteria
3. If the answer to 2. was “Yes, some HVAs meet the criteria” or “No”, name which criteria are not met	Pick from the list: arise from the pleural line; moves with lung sliding; extends to the bottom of the screen
4. Is the number of HVAs easy to count?	1—easy; 2—somewhat challenging; 3—very difficult; 4—not able to assess
5. If the answer is 3 or 4, explain why	Pick from the list: the edges are not clear; artifacts blend; too grainy; move too quickly; variable width. Also, count HVAs that reach the bottom of the screen and ones that don’t.
6. Assign a LUSS score for HVAs †	0—≤2HVAs; 1—≤50% pleural line occupied by HVAs; 2—50–100% pleural line occupied by HVAs
7. Rate the overall quality of the image with regard to the ability to assess B-lines on a scale from 1 to 100	Open question
8. What is the reason for such assessment, name the features of artifacts that are influencing your judgement.	Pick from the list: the edges are not clear; artifacts blend; too grainy; variable width; variable vertical echogenicity
Part 2: the comparison of paired clips	
1. In your opinion are these two clips of equal quality?	Yes/No
2. If No, which one do you consider higher quality?	Left/Right
3. Why do you think the quality of that image is better?	Mark one or more answers: Contrast between vertical artifact and surrounding tissue is greater The movement of vertical artifacts is easier to assess Vertical artifacts are easier to count The pleural line quality is easier to assess
4. Is the quality of B-lines equal on both clips?	Yes/No
5. If No, where is the quality of B-lines better?	Left/Right

* The criteria for B-lines were defined based on guidelines from human medicine. The “true” B-line had to arise from the pleural line, move with lung sliding, and extend to the bottom of the screen irrespective of imaging depth. † The LUSS score was adapted from the study of Oricco et al. [18].

**Table 2 vetsci-12-00949-t002:** Concordance coefficients for the assessment of transducer characteristics (N = 24).

			95% CI	
Parameter	Concordance Coefficient	*p*	LL	UL
Are HVAs visible?	1.00 ^a^	<0.001	1.00	1.00
Do HVAs meet the criteria of B-lines?	0.59 ^b^	0.002	0.38	0.75
Not met: HVAs arise from the pleural line	−0.06 ^a^	0.758	−0.17	0.00
Not met: HVAs move with lung sliding	0.75 ^a^	<0.001	0.45	1.00
Not met: HVAs extend to the bottom of the screen	0.58 ^a^	0.003	0.22	0.90
Are HVAs easy to count?	0.74 ^b^	<0.001	0.53	0.86
HVA edges not clear	0.51 ^a^	0.008	0.13	0.79
HVAs blend	0.51 ^a^	0.008	0.15	0.75
HVAs grainy	0.83 ^a^	<0.001	0.58	1.00
HVAs move too quickly	0.07 ^a^	0.729	−0.29	0.51
HVAs of variable width	0.43 ^a^	0.032	0.00	0.75
Number of HVAs reaching the bottom of the screen	0.58 ^c^	0.003	0.15	0.79
Number of HVAs not reaching the bottom of the screen	0.87 ^c^	<0.001	0.49	0.97
LUSS value	0.90 ^b^	<0.001	0.73	0.98
Overall image quality scale	0.82 ^c^	<0.001	0.66	0.93
HVA edges not clear	0.37 ^a^	0.058	0.02	0.75
HVAs blend	0.58 ^a^	0.002	0.27	0.87
HVAs grainy	0.35 ^a^	0.072	−0.05	0.79
HVAs of variable width	0.26 ^a^	0.202	−0.15	0.57
HVAs of variable echogenicity	0.41 ^a^	0.045	0.00	0.75
Pair: equal image quality	0.63 ^a^	<0.001	0.00	1.00
Image of higher quality	0.80 ^a^	<0.001	0.49	1.00
Superior HVA echogenicity	1.00 ^a^	<0.001	1.00	1.00
HVA movement easier to assess	0.05 ^a^	0.786	−0.31	0.42
HVAs easier to count	0.72 ^a^	<0.001	0.46	1.00
Pleural line easier to assess	0.90 ^a^	<0.001	0.60	1.00
Pleural line abnormalities more visible	0.78 ^a^	<0.001	0.00	1.00
Pair: equal B-line quality	0.83 ^a^	<0.001	0.00	1.00
Image with higher B-line quality	0.79 ^a^	<0.001	0.47	1.00

^a^ Kappa coefficient of agreement for nominal scale parameters; ^b^ Kendall’s tau coefficient of agreement for ordinal scale parameters; ^c^ Pearson correlation coefficient for quantitative scale parameters. CI—confidence interval, HVAs—hyperechoic vertical artifacts, LL—lower confidence limit, LUSS—lung ultrasound score, UL—upper confidence limit.

**Table 3 vetsci-12-00949-t003:** Cine-loop comparison for diagnostic characteristics of microconvex, phased array, and linear transducers.

	Transducer Type			
	Microconvex (*N* = 16)	PA (*N* = 16)	Linear (*N* = 16)	
	M ± SD/*n* (%)	M ± SD/*n* (%)	M ± SD/*n* (%)	*p*
Are HVAs visible? (+)	16 (100.0%)	16 (100.0%)	16 (100.0%)	1.000
Do HVAs meet the criteria of B-lines?				0.024
Yes, all meet	8 (50.0%) ^AB^	12 (75.0%) ^B^	3 (18.8%) ^A^	
Yes, some meet	7 (43.8%)	3 (18.8%)	9 (56.3%)	
No	1 (6.3%)	1 (6.3%)	4 (24.0%)	
Arise from the pleural line (+)	0 (0.0%)	1 (6.3%)	2 (12.5%)	0.344
Move with lung sliding (+)	4 (25.0%)	2 (12.5%)	4 (25.0%)	0.603
Extend to the bottom of the screen (+)	8 (50.0%) ^AB^	3 (18.8%) ^A^	10 (62.5%) ^B^	0.037
HVAs easy to count				<0.001
Easy	0 (0.0%) ^A^	0 (0.0%) ^A^	6 (35.7%) ^B^	
Somewhat challenging	8 (50.0%)	2 (12.5%)	6 (35.7%)	
Very difficult	6 (37.5%)	9 (56.3%)	4 (25.0%)	
Not able to assess	2 (12.5%) ^AB^	5 (31.3%) ^B^	0 (0.0%) ^A^	
Edges not clear_1 (+)	9 (56.3%) ^B^	15 (93.8%) ^C^	2 (12.5%) ^A^	<0.001
HVAs blend (+)	6 (37.5%) ^A^	13 (81.3%) ^B^	7 (43.8%) ^AB^	0.027
HVAs grainy (+)	3 (18.8%) ^A^	16 (100.0%) ^B^	1 (6.4%) ^A^	<0.001
Move too quickly (+)	6 (37.5%)	10 (62.5%)	5 (31.3%)	0.169
Variable width (+)	7 (43.8%)	11 (68.8%)	5 (31.3%)	0.097
HVAs that reach the bottom of the screen	6.72 ± 7.03	3.91 ± 4.36	3.88 ± 3.69	0.389
HVAs that do not reach the bottom of the screen	0.94 ± 1.18 ^A^	0.28 ± 0.89 ^A^	3.79 ± 3.64 ^B^	<0.001
LUSS value				0.370
<25%	2 (12.5%)	4 (25.0%)	5 (31.3%)	
<50%	6 (37.5%)	2 (12.5%)	5 (31.3%)	
50–100%	8 (50.0%)	10 (62.5%)	6 (37.5%)	
Overall image quality	77.81 ± 10.16 ^B^	32.19 ± 25.95 ^A^	80.00 ± 10.80 ^B^	<0.001
HVA edges not clear (+)	11 (68.8%) ^B^	16 (100.0%) ^C^	4 (25.0%) ^A^	<0.001
HVAs blend (+)	9 (56.3%) ^AB^	14 (87.5%) ^B^	6 (37.5%) ^A^	0.014
HVAs too grainy (+)	5 (31.3%) ^A^	16 (100.0%) ^B^	5 (31.3%) ^A^	<0.001
Variable width (+)	12 (75.0%)	13 (81.3%)	7 (43.8%)	0.055
Variable echogenicity (+)	10 (62.5%)	7 (43.8%)	10 (62.5%)	0.467

^ABC^ Different superscript letters indicate transducer types whose column proportions differ significantly from each other at the *p* < 0.05 level. HVAs, hyperechoic vertical artifacts; PA, phased array.

**Table 4 vetsci-12-00949-t004:** Image quality analysis for paired cine-loops from the microconvex, phased array, and linear transducers.

Dependent Variable	Microconvex (*n* = 16)	PA (*n* = 16)	Linear (*n* = 16)	*F*	η^2^
	*M* ± *SD*	*M* ± *SD*	*M* ± *SD*		
Better quality	1.31 ± 0.36 ^B^	0.00 ± 0.00 ^C^	1.69 ± 0.36 ^A^	145.89 ***	0.87
Superior HVA echogenicity	1.22 ± 0.26 ^B^	0.00 ± 0.00 ^C^	1.78 ± 0.26 ^A^	303.21 ***	0.93
HVA movement easier to assess	1.28 ± 0.45 ^A^	0.78 ± 0.41 ^B^	0.94 ± 0.36 ^AB^	6.36 **	0.22
HVAs easier to count	1.31 ± 0.36 ^B^	0.03 ± 0.13 ^C^	1.66 ± 0.35 ^A^	130.99 ***	0.85
Pleural line quality easier to assess	1.22 ± 0.26 ^B^	0.00 ± 0.00 ^C^	1.78 ± 0.26 ^A^	303.21 ***	0.93
Higher HVA quality	1.22 ± 0.36 ^B^	0.03 ± 0.13 ^C^	1.75 ± 0.32 ^A^	149.94 ***	0.87

*** *p* < 0.001; ** *p* < 0.010. *Note:* Various letter indices mark values with significant statistical differences at the *p* < 0.05 level. HVAs, hyperechoic vertical artifacts; PA, phased array.

## Data Availability

The original contributions presented in this study are included in the article. Further inquiries can be directed to the corresponding author(s).

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
