# Peer review of "Sonographic Assessment of Hyperechoic Vertical Artifact Characteristics in Lung Ultrasound Using Microconvex, Phased Array, and Linear Transducers"

_vetsci, 2025, doi:10.3390/vetsci12100949_

Round 1
Reviewer 1 Report
Comments and Suggestions for Authors
This study inquiries about the effect of ultrasound probes in the classification and description of HVA in LUS. The study is correctly written and presented. The conclusions are supported by the results, and the methodology is correctly described.
I have some comments to make about the whole work and some particular points:
- I really miss some representative figures showing the described differences in between images from the different US probes. From my feeling, a paper related to ultrasonography must have at least a couple of representatives images to exemplify the text.
- Regarding the US examination protocol, I have a couple of comments:
- I would like to have a short description of the scanning protocol (line 76, references 15 and 16). Few lines explaining how the authors made the scans. Briefly, that’s enough.
- Here I have my biggest doubt in this paper. Knowing that the US probes had significant differences in frequency, which leads to a difference in ultrasound penetration, I don’t understand why the authors used a fixed field of view of 8-10cm (scan depth). It’s obvious that the linear transducer is going to have problems in visualizing the structures at 10cm, it is just physics. This fact makes me doubt the concept of B-lines reaching the bottom of the screen as a criterion for classifying the lines as B-lines. And the situation is worse when the authors fixed the TCG instead of adjusting the gains by level to compensate for this frequency difference between probes. I think there is a problem in the examination settings for this particular concept (lines reaching the bottom of the image). An example image of the scan which each probe could help to confirm or reject this hypothesis.
- In table 1, the questions about the quality of HVA, in number 7, I’m curious why you set the scale from 1 to 100 for the overall quality. Why not 1 to 10? These huge range can create more confusion than accuracy in the statistics.
- In table 4, what is the meaning of each parameter in the dependent variable row? Please include it at the foot of the table.
- The discussion part is fine, but the extension of the lines to the bottom of the screen is affected, in my honest opinion, by the significant difference in frequency between probes. As the authors described and correctly reflected in the text, there is a difference between PA and linear probes in terms of lines reaching the bottom of the screen. I don’t think it is because of the quality of the images but the different penetration of the ultrasound waves in the tissues.
- I would rewrite the text from lines 234 to 248. All this paragraph is a bit messy and hard to read.
In summary, nice work. Congratulations to the authors. But I think there is a potential error in the scanning protocol that affects one of the parameters used for B-line classification.
Author Response
We would like to thank the reviewer for their positive feedback of this manuscript.
I have some comments to make about the whole work and some particular points:
- I really miss some representative figures showing the described differences in between images from the different US probes. From my feeling, a paper related to ultrasonography must have at least a couple of representatives images to exemplify the text.
Thank you for the suggestion. We agree and have added figure 1 representing the images from various transducers.
Figure legend: “Figure 1. Still images from the cine-loops recorded with three transducer types (from left to right): microconvex, PA, and linear.”
- Regarding the US examination protocol, I have a couple of comments:
- I would like to have a short description of the scanning protocol (line 76, references 15 and 16). Few lines explaining how the authors made the scans. Briefly, that’s enough.
We appreciate the request from the reviewer and have added this brief description into the materials and methods section.
Lines 76-83: “The LUS technique was performed according to a sliding protocol described elsewhere15,16. Briefly, it utilizes a horizontal sliding technique with the transducer placed at three different vertical locations on each side of the thorax. The starting orientation of the probe is perpendicular to the ribs. The transducer is slid along a dorsal line from cranial to caudal, rotated 90 degrees into a transverse plane (parallel to the ribs), and slid again from caudal to cranial direction along the same line. The process is then repeated in the middle and ventral thirds of the hemithorax, and then repeated on the contralateral hemithorax
- Here I have my biggest doubt in this paper. Knowing that the US probes had significant differences in frequency, which leads to a difference in ultrasound penetration, I don’t understand why the authors used a fixed field of view of 8-10cm (scan depth). It’s obvious that the linear transducer is going to have problems in visualizing the structures at 10cm, it is just physics. This fact makes me doubt the concept of B-lines reaching the bottom of the screen as a criterion for classifying the lines as B-lines. And the situation is worse when the authors fixed the TCG instead of adjusting the gains by level to compensate for this frequency difference between probes. I think there is a problem in the examination settings for this particular concept (lines reaching the bottom of the image). An example image of the scan which each probe could help to confirm or reject this hypothesis.
We thank the reviewer for this thoughtful comment and have clarified these concerns in the manuscript in the materials and methods, the discussion, and the limitations. The authors understand the concern raised by the reviewer and agree with the possible impact this could have on the interpretation of vertical artefacts and their classification as B- or other lines. That said, there is a lack of consensus regarding the definition of B-lines in the literature, particularly with regard to a B-line having to extend to the far field of the image, which holds true even with different probe types. This variation, particularly with changes in the probe used, was a big reason we wanted to undertake the study, to draw attention to the matter within the veterinary profession.
It is also the authors' experience that veterinary practitioners often scan the lungs without optimization images for the different probes; failing to adjust the depth and TGC, for example. Although we agree with the reviewer that this can impact the findings and the visualization of HVAs, we wanted the study to reflect veterinary clinical reality. We therefore elected to perform this study without modifying the settings to better study the impact this would have in a clinical setting. We agree that many of the findings might be explained by these unmodified machine settings in relation to the patient, the structure to be visualized, and the probe. We have clarified this in the discussion and limitation section.
Regarding the current study, we have chosen to include the criteria that B-lines reach the far field of the screen, as that was the original definition of a B-line published in the early works of Daniel Lichtenstein.( Lichtenstein DA. Lung Ultrasound in the Critically III. London Springer 2016;376.)
Regarding the concept of setting the imaging depth to 8-10 cm, we chose this to highlight to readers that HVAs are artifacts arising from the lung surface and not lung tissue, despite how far they may or may not penetrate into the far field. By extending the depth to 8-10 cm we hoped to highlight the fact there is a difference between probes, that may or may not be correctable with a change in machine settings, and if one follows the doctrine that B-lines must reach the far field (a fact not all the authors on this paper are fully committed to) then it becomes evident that interpretation can be impacted by a change in the probe used (with or without optimizing the image for that probe). We agree with the reviewer that the characteristics and design of the linear probe suggest that it should be used for the assessment of the pleural line and superficially located structures. In contrast, phased array and microconvex probes are intended for imaging deeper structures. Artifacts (i.e., HVAs) arise from the pleural line, highlighting to readers that we are not observing tissue, only acoustic artifact from the interaction of the ultrasound beam with the lung surface and air. Therefore, contrary to popular belief, we believe a greater imaging depth can also be set for the linear transducer, as briefly explained in the text.
Line 218-222: “. In our study, TGC was set to a mid-range level throughout the imaging depth. This was done to mimic a busy general practice or emergency clinical setting where there may not be time to adjust the TGC scale when performing rapid sonographic lung evaluations in dyspneic patients. Furthermore, in the GE Vivid IQ machine, the TGS is set in the same manner for the microconvex and PA transducers, and the intention was to keep the same machine settings for all transducers. It is possible that the appearance and length of the HVAs would have changed, and assessment by the experts may have been different had the TGC been optimized throughout the field of vision for each probe and for each animal.”
Additional studies should clarify to what extent our findings are explained by the probe itself or by the frequency and TCG settings. This information will be pivotal for general veterinary practitioners to understand the impact (or lack of) modified machine settings on the images obtained.
- In table 1, the questions about the quality of HVA, in number 7, I’m curious why you set the scale from 1 to 100 for the overall quality. Why not 1 to 10? These huge range can create more confusion than accuracy in the statistics.
Thank you for this comment. There was no specific rationale for using a 1-100 scale vs a 1-10 scale. And the scale had a high concordance coefficient. Whether that would be different had the scale been set to 1-10 remains unknown.
- In table 4, what is the meaning of each parameter in the dependent variable row? Please include it at the foot of the table.
Thank you, we have corrected the table for clarity, and added the descriptions in the footnote. (LL = lower confidence limit; UL = upper confidence limit)
- The discussion part is fine, but the extension of the lines to the bottom of the screen is affected, in my honest opinion, by the significant difference in frequency between probes. As the authors described and correctly reflected in the text, there is a difference between PA and linear probes in terms of lines reaching the bottom of the screen. I don’t think it is because of the quality of the images but the different penetration of the ultrasound waves in the tissues.
We thank the reviewer and agree different probes have different penetration depths and crystal arrangements. Therefore, we believe they cannot be used interchangeably, which was the rationale for the present study. However, in lung ultrasonography, the settings of the ultrasound system are of fundamental importance, which has been demonstrated in numerous phantom studies. This is related not only to the type of probe but also to the focal setting and the activation of harmonic imaging (but that is beyond the scope of this study).
We think this is a concern, since in lung ultrasonography, clinicians use linear, convex, and phased array probes interchangeably. While the linear probe is designed for the assessment of superficial structures, hence the focus of the acoustic lens is located close to the probe surface, the convex and phased array probes are intended for imaging deeper structures, which results in the acoustic lens focus being positioned much farther from the probe surface. (https://doi.org/10.1016/j.ultrasmedbio.2019.03.003)
We agree that the highest frequency is most commonly used in ultrasonography; however, LUS is arguably an exception to this rule. The recommendations regarding the use of the lowest probe frequency arise from several factors. Firstly, this is related to sonic channels and acoustic traps that generate artifacts. Artifact length: an HVA may start at the pleural line and extend to the bottom of the screen. The length depends on the time required for the acoustic trap to re-emit the energy of the pulse that has been captured. Thus, the length depends on the shape of the trap, but also on machine settings such as Time Gain Compensation (TGC), focal position, or probe frequency adjustment. When changing the frequency, for example, on a GE linear probe 6–12 MHz, the artifacts shorten. Similarly, in the article by Buda et al. (Clinical Impact of Vertical Artifacts Changing with Frequency in Lung Ultrasound), changing the probe frequency affected the artifact length from B to Z, as higher frequencies are attenuated more rapidly. The same applies to articles describing ultrasound machine settings during lung ultrasonography. (https://doi.org/10.1016/j.ultrasmedbio.2019.03.003)
I would rewrite the text from lines 234 to 248. All this paragraph is a bit messy and hard to read
The manuscript has been rewritten for clarity. The paragraph now reads:
Compared with the linear transducer, it was harder to clearly distinguish individual HVAs with the PA transducer. Furthermore, there was no consensus among evaluators regarding whether HVAs originated from the pleural line, and only limited agreement on whether they extended to the bottom of the screen. These inconsistencies may have led to only moderate agreement regarding the fulfillment of B-line criteria. This finding underscores that B-line assessment is more complex than suggested by prior studies. In contrast, interrater agreement was high for quantitative parameters, including assess-ment if HVAs did or did not reach the bottom of the screen and overall image quality. These results indicate that structured numerical scales, such as the LUSS score or Likert-based measures with predefined criteria, may reduce interrater variability, whereas subjective parameters remain less reliable.
In summary, nice work. Congratulations to the authors. But I think there is a potential error in the scanning protocol that affects one of the parameters used for B-line classification.
We hope we have provided sufficient information to clarify the reviewer concerns. We believe the addition of this information to the materials and methods, discussion, and limitations has significantly improved the manuscript and thank the reviewer for their contribution.

Reviewer 2 Report
Comments and Suggestions for Authors
This article addresses a significant gap in veterinary lung ultrasound by comparing various types of transducers for assessing HVAs in dogs with presumed (? It’s not clear) lung pathology. The argument is current, citing the existence of contrastive veterinary studies and human consent guidelines. The main points include a standardized process, the scientific evaluation by the reviewers, and statistical rigor. However, the work has some noteworthy flaws, including grammatical and orthographic errors, methodological errors (e.g., revisor bias, low sample), non-reportable limitations (e.g., lacking of gold standard), scientific inconsistencies (e.g., definitions of line B that are inconsistent with literature), and bibliographic issues.
Below, I detail errors and issues categorically, referencing page/line numbers from the provided PDF transcript
The abstract and the text are inconsistent; in some places, it states that there are 24 cinecycles, while in other places, it states that there are 16. Verify and make corrections.
Line 18: Put "that" after "Let's assume"
Line 25: Do you think the term Pearson's tau is appropriate? Verify.
lines 31–32. In percentage values, use a period for the comma. Go over the entire text.
The phrase "companion animals" should be avoided as the study was only done on dogs, according to line 54. Kindly revise the text.
Line 60: To be accurate, the paragraph should be titled 2.1 rather than 3. 1.
Line 61: Regardless of the cause, those who had vertical artifacts who came in for cardiac examination were employed, according to paragraph 3.1. The appearance of vertical artifacts might vary based on the underlying diseases. Comments on this ought to be made in the discussions area.
Line 64: Kindly provide the file number for the local ethics committee.
Line 116: Oricco et al. is listed as the reference, however it is not included in the list of references. Verify.
Line 125: Krippendorf's use is noted, although the findings do not cite him. Verify.
Line 141: What LL and UL are is not stated in Table 2's legend. "HVA edges not clear" is also noted twice. Please edit and specify.
Line 154: "Transducer" appears more than once. Examine and remove.
Line 157: "Do HVAs meet the criteria of B lines" There is no question mark.
Line 167: Since it is commonly advised to utilize maximum frequencies, the discussions do not address how the settings used, in particular the lowest frequency chosen for each probe, could have affected the results.
Line 191 reads: "The machine settings used in the current study were standardized for all transducers to minimize any impact on image quality," according to remark line 167. I disagree that the settings were optimal for obtaining high-quality images. Every probe's frequency should always be set to its maximum rather than its minimum. At least that's what I think. Please rewrite this sentence.
Line 222: The given statement is not accurate. According to references 7 and 18, Z and I lines also seem to have a pathological origin. Please rewrite.
Line 239: Although experts concurred that the B-line requirements were satisfied, they couldn't agree on the rationale for their evaluation. This sentence need revision or more insightful commentary. The parameters and criteria used to define the b-lines may also have an impact on this. The fact that two reviewers disagree on this is unbelievable to me.
Line 288, change III to Ill.
Line 297: Add the missing article number.
Line 314: Add the missing article number.
Line 316: An author and the article number are missing.
Line 318: Add the missing item number.
Check that all references have 3 authors followed by et al.
Comment about how the results may have been affected if the TGC had been deeper.
Discuss about how the results could have been affected by the limited number of cases.
It is also hard to determine whether B-lines or other vertical artifacts that are similar to B-lines but have somewhat different features because they are not diseased were being analyzed because the exact pathologies of the individuals included in the study were not disclosed clearly. Contribute in discussions by adding a comment.
For statistical comparisons, Bonferroni is absent. Assess and add if required.
The generalization is inaccurate because there is no mention of the fact that only one machine was utilized. Please add a comment
None
Author Response
This article addresses a significant gap in veterinary lung ultrasound by comparing various types of transducers for assessing HVAs in dogs with presumed (? It’s not clear) lung pathology. The argument is current, citing the existence of contrastive veterinary studies and human consent guidelines. The main points include a standardized process, the scientific evaluation by the reviewers, and statistical rigor. However, the work has some noteworthy flaws, including grammatical and orthographic errors, methodological errors (e.g., revisor bias, low sample), non-reportable limitations (e.g., lacking of gold standard), scientific inconsistencies (e.g., definitions of line B that are inconsistent with literature), and bibliographic issues.
Below, I detail errors and issues categorically, referencing page/line numbers from the provided PDF transcript
We thank the reviewer for their appraisal of our manuscript, and hope we have addressed all of the concerns raised by the reviewer in the following detailed responses.
The abstract and the text are inconsistent; in some places, it states that there are 24 cinecycles, while in other places, it states that there are 16. Verify and make corrections.
Thank you for pointing this out. We overlooked the lack of patient number in the materials and methods section. We have corrected that mistake.
We had 8 dogs and 3 transducers. This makes 24 cine-loops, which were assessed by 2 reviewers. So for concordance, we had 24 cine-loops. When image quality was evaluated on individual clips and paired clips, we had 8 clips from each probe assessed by 2 reviewers: 8 x 2 = 16.
Line 18: Put "that" after "Let's assume"
We apologize, but we can't find the phrase "let's assume" anywhere in the paper. Is there another place the reviewer wanted us to add the word "that"? Please clarify.
Line 25: Do you think the term Pearson's tau is appropriate? Verify.
Thank you for noting the discrepancy. We have removed "tau" from the description of the statistics used. The sentence now reads:
Interrater concordance was determined using the Kappa coefficient, Kendall's tau, and Pearson correlation coefficient.
lines 31–32. In percentage values, use a period for the comma. Go over the entire text.
The manuscript has been adjusted accordingly.
The phrase "companion animals" should be avoided as the study was only done on dogs, according to line 54. Kindly revise the text.
The manuscript has been modified accordingly.
Line 60: To be accurate, the paragraph should be titled 2.1 rather than 3. 1.
We thank the reviewer for having spotted this error. The manuscript has been adapted accordingly.
Line 61: Regardless of the cause, those who had vertical artifacts who came in for cardiac examination were employed, according to paragraph 3.1. The appearance of vertical artifacts might vary based on the underlying diseases. Comments on this ought to be made in the discussions area.
We agree with the reviewer that the appearance of vertical artefacts might vary based on the underlying disease. That said, as HVAs evaluated within each animal should be the same for each probe used, any differences between probes should not have been due to different disease processes. On a bigger scale, beyond individual animals, because animals were referred for cardiac work up, it is true most animals likely had left sided congestive heart failure, and if a wider range of dogs with a greater variation in underlying pathology (or even all dogs with a specific pathology such as pulmonary fibrosis) were included, the impact of the probe type on the appearance of B-line, and how they were assessed could have been different. We have added a line to the limitations section to highlight this fact. The sentence now reads: "The study population was small and focused on dogs presenting for cardiac consultation. It is unknown if results would be repeatable using a larger group and/or including dogs with a wider range of underlying lung pathologies".
Line 64: Kindly provide the file number for the local ethics committee. The local ethics committee didn’t recognize this study as an experiment, and therefore did not provide an ethics file number. Their rationale was the following: the study does not consider the research as an 'animal experiment' in legal terms, indicating that this was a harmless procedure that did not require thorough ethical review and registration of the data. Hence, we cannot provide a number. We have stated this in the methods section of the paper.
More specifically, the local ethics committee replied as follows:
Pursuant to Article 1(1) of the Act of 15 January 2015 on the Protection of Animals Used for Scientific or Educational Purposes (hereinafter referred to as the "Act"), it sets out, among other things, the rules for carrying out procedures and conducting experiments, as well as the conditions for keeping animals used for scientific or educational purposes, and the manner of handling such animals.At the same time, the provisions of the Act do not apply to veterinary services as defined in the Act of 18 December 2003 on Veterinary Treatment Facilities, nor to activities which, in accordance with veterinary medical practice, do not cause the animal pain, suffering, distress, or lasting harm at a level equivalent to, or greater than, that caused by the introduction of a needle, as specified in Article 1(2)(1) and (5) of the Act.
As mentioned above, the LKE grants consent for the conduct of experiments, which, according to Article 2(1)(7) of the Act, are research programs involving a procedure or procedures with a specified scientific or educational objective. Consent thus covers a range of procedures which, according to Article 2(1)(6) of the Act, involve any use of animals for the purposes specified in Article 3 that may cause the animal pain, suffering, distress, or lasting harm at a level equivalent to, or greater than, that caused by the introduction of a needle.
In view of the above, in the opinion of the First Local Ethical Committee (I LKE) in Warsaw, the examination you have described does not constitute an experiment within the meaning of the Act. In particular, it cannot be concluded that any procedures are involved in the above-mentioned diagnostic laboratory examination. Furthermore, diagnostic laboratory examinations fall under the exemption referred to in Article 1(2)(1) of the Act, as they constitute laboratory services within the meaning of the Act on Veterinary Treatment Facilities.
Line 116: Oricco et al. is listed as the reference, however it is not included in the list of references. Verify.
Thank you for noting the oversight. The reference has been added.
Line 125: Krippendorf's use is noted, although the findings do not cite him. Verify.
We thank the reviewer for having spotted this. The values are not mentioned in the table, but they have been added to the “Results” section, which now reads:
In summary, Krippendorff's α coefficient was used to verify the overall agreement be-tween experts for parameters assessed on nominal, ordinal, and quantitative scales. The results indicated that the agreement for 23 parameters assessed on a nominal scale was moderate (α = 0.68; 95% CI: LL = 0.61; UL = 0.75), the agreement for 3 parameters assessed on an ordinal scale was good (α = 0.79; 95% CI: LL = 0.65; UL = 0.90), and the agreement for 3 parameters assessed on a quantitative scale was very good (α = 0.95; 95% CI: LL = 0.88; UL = 0.99).
Line 141: What LL and UL are is not stated in Table 2's legend. "HVA edges not clear" is also noted twice. Please edit and specify.
Thank you for the comment and for noting the missing abbreviations. They have been added.
LL = lower confidence limit
UL = upper confidence limit
The use of "HVA edges not clear" was not used erroneously, but intentionally included twice for reviewers to interpret in different parts of the assessment– once in the evaluation of ease/difficulty in counting HVAs, and again in the reasons for the image quality value on the Likert scale. Hopefully that makes sense.
Line 154: "Transducer" appears more than once. Examine and remove.
The manuscript has been adapted accordingly.
Line 157: "Do HVAs meet the criteria of B lines" There is no question mark.
The manuscript has been adapted accordingly.
Line 167: Since it is commonly advised to utilize maximum frequencies, the discussions do not address how the settings used, in particular the lowest frequency chosen for each probe, could have affected the results.
We thank the reviewer for this thoughtful comment, which the other reviewer has also raised.
We agree that the highest frequency is most commonly used in ultrasonography; however, LUS is arguably an exception to this rule. The recommendations regarding the use of the lowest probe frequency arise from several factors. Firstly, this is related to sonic channels and acoustic traps that generate artifacts. Artifact length: an HVA may start at the pleural line and extend to the bottom of the screen. The length depends on the time required for the acoustic trap to re-emit the energy of the pulse that has been captured. Thus, the length depends on the shape of the trap, but also on machine settings such as Time Gain Compensation (TGC), focal position, or probe frequency adjustment. When changing the frequency, for example, on a GE linear probe 6–12 MHz, the artifacts shorten. Similarly, in the article by Buda et al. (Clinical Impact of Vertical Artifacts Changing with Frequency in Lung Ultrasound), changing the probe frequency affected the artifact length from B to Z, as higher frequencies are attenuated more rapidly. The same applies to articles describing ultrasound machine settings during lung ultrasonography. (https://doi.org/10.1016/j.ultrasmedbio.2019.03.003)
Line 191 reads: "The machine settings used in the current study were standardized for all transducers to minimize any impact on image quality," according to remark line 167. I disagree that the settings were optimal for obtaining high-quality images. Every probe's frequency should always be set to its maximum rather than its minimum. At least that's what I think. Please rewrite this sentence.
Thank you for this comment. Yes, image quality increases with higher frequencies when assessing tissues. As we discussed above, lung ultrasound is based on the generated artifacts, and HVAs tend to shorten with higher frequencies (reference 18). The sentence was rewritten and now reads:
The machine settings used in the current study were standardized for all transducers to minimize any variability in image quality and HVA characteristics.
Line 222: The given statement is not accurate. According to references 7 and 18, Z and I lines also seem to have a pathological origin. Please rewrite.
The reviewer brings up a very interesting point, highlighting some of the controversy and lack of consensus in both the human and veterinary professions. There are references alluding to Z-lines and I-lines not having clinical significance, while others may suggest they do. Looking more closely at reference 18, Buda et al state "Z-line artifacts have not yet been described as having clinical significance", which further adds to the lack of a clear pathology. Given there is also controversy regarding the definition of Z and I-lines (which was discussed in the manuscript), we have chosen not to go into great depth on the origins of Z and I-lines as we feel it will detract from the objective of the study, and we find a consensus on their clinical significance challenging to pin. If the reviewer feels strongly and can provide clear statements with references that allow us to discuss the origin and pathology of Z and I-line in more detail, we can incorporate a longer discussion on the subject. However, we worry this will add more confusion than clarity for readers and detract from the main study objectives. We did, however, modify the sentence to be less "biased" in one direction vs. another. The sentence now reads: "From a clinical perspective, however, the most important issue is to distinguish “true” B-lines from other HVAs (regardless of the adopted nomenclature), because B-lines are considered pathological artifacts, whereas other HVAs, depending on the author, are reported to have variable clinical significance and lack a clearly defined pathological basis.
Line 239: Although experts concurred that the B-line requirements were satisfied, they couldn't agree on the rationale for their evaluation. This sentence need revision or more insightful commentary. The parameters and criteria used to define the b-lines may also have an impact on this. The fact that two reviewers disagree on this is unbelievable to me.
Thank you for this commentary. We have rewritten these sentences to clarify our thoughts. The paragraph now reads:
Compared with the linear transducer, it was harder to clearly distinguish individual HVAs with the PA transducer. Furthermore, there was no consensus among evaluators regarding whether HVAs originated from the pleural line, and only limited agreement on whether they extended to the bottom of the screen. These inconsistencies may have led to only moderate agreement regarding the fulfillment of B-line criteria. This finding underscores that B-line assessment is more complex than suggested by prior studies. In contrast, interrater agreement was high for quantitative parameters, including assess-ment if HVAs did or did not reach the bottom of the screen and overall image quality. These results indicate that structured numerical scales, such as the LUSS score or Likert-based measures with predefined criteria, may reduce interrater variability, whereas subjective parameters remain less reliable.
Line 288, change III to Ill.
The manuscript has been adapted accordingly.
Line 297: Add the missing article number.
Line 314: Add the missing article number.
Line 316: An author and the article number are missing.
Line 318: Add the missing item number.
Check that all references have 3 authors followed by et al.
The manuscript has been adapted accordingly.
Comment about how the results may have been affected if the TGC had been deeper.
The section on study limitations now reads:
The study population was small and focused to dogs presenting for cardiac consultation. It is unknown if results would be repeatable using a larger group and/or including dogs with a wider range of underlying lung pathologies. The TGC was not adjusted to the depth of imaging on the linear probe which may have impacted the visualization of HVAs as discussed above. All examinations were performed with only one type of ultrasound machine, and image quality may differ on other machines. Lastly, the evaluation of the cine-loops was retrospective, so the perceived image quality may have differed if the assessment was performed in real time. The quality loss due to the transfer and conversion of the files was probably minimal and equal across all cine-loops, given that only one machine was used for all image captures.
Discuss about how the results could have been affected by the limited number of cases.
As above.
It is also hard to determine whether B-lines or other vertical artifacts that are similar to B-lines but have somewhat different features because they are not diseased were being analyzed because the exact pathologies of the individuals included in the study were not disclosed clearly. Contribute in discussions by adding a comment.
Thank you for this comment. Given the objective of the study was to assess HVAs regardless of the pathology, we did not design the study to assess if the disease states could impact the appearance of B-lines (that is another branch of research we are also undertaking, but felt the current study needed to be addressed before moving to the next stage). As we acknowledge underlying lung pathologies may results in different HVA characteristics we have added a section to the limitations and the discussion to address this. Given the impact of different pathologies on the appearance of HVAs is in it's infancy, we are not sure how much can be added to the discussion at this point and hope the reviewer finds the changes added sufficient to address their concerns.
For statistical comparisons, Bonferroni is absent. Assess and add if required.
Thank you for this insight. We have chosen other statistical tests for concordance, and for paired clip comparisons, we used a Games-Howell post-hoc test. This decision was made with the statisticians.
The generalization is inaccurate because there is no mention of the fact that only one machine was utilized. Please add a comment
Thank you for this comment. We have added a sentence mentioning the use of only one machine as a limitation of the study.
The sentence now reads: All examinations were performed with only one type of ultrasound machine, and image quality may differ on other machines.

Round 2
Reviewer 1 Report
Comments and Suggestions for Authors
The authors addressed all my comments with clarity and showing a deep knowledge in the field.
Reviewer 2 Report
Comments and Suggestions for Authors
Line 202: I appreciate the author's response to my previous comment. Care should be taken when using the findings of one or more human studies. In my opinion, the convex probe scans of elderly heart failure patients in humans are not comparable to the microconvex probe scans of our canine patients with the same problem.
The first bias discussed in the cited paper is that the results could have been impacted by external factors such internal structure, lateral width, and brightness. Furthermore, as most studies are carried out with high frequencies, I believe it is appropriate to mention in discussions that a higher frequency wasn't used and that it is unknown how this could have affected the results.